# Association of Sleep Quality and Macronutrient Distribution: A Systematic Review and Meta-Regression

**DOI:** 10.3390/nu12010126

**Published:** 2020-01-02

**Authors:** Clarinda Nataria Sutanto, Min Xian Wang, Denise Tan, Jung Eun Kim

**Affiliations:** Food Science and Technology Programme, Department of Chemistry, National University of Singapore, 3 Science Drive 3, Singapore 117543, Singapore; E0254848@u.nus.edu (C.N.S.); ephwmx@nus.edu.sg (M.X.W.); E0384124@u.nus.edu (D.T.)

**Keywords:** sleep quality, macronutrients distribution, dietary protein, acceptable macronutrient distribution range

## Abstract

Sleep is involved in metabolic, emotional and cognitive regulation and is therefore an essential part of our health. Although an association between sleep quality and macronutrient intake has been reported, studies on the effect of macronutrient distribution with sleep quality are limited, and available results are inconsistent. In this study, we aim to assess the association between sleep quality and macronutrient distribution in healthy adults from systematically reviewed cross-sectional studies and randomized controlled trials (RCTs). A total of 19 relevant articles were selected and it was observed that good sleepers (sleep duration ≥ 7 h, global sleep score ≤ 5, sleep latency ≤ 30 min and sleep efficiency >85%) had a higher energy distribution from dietary protein than poor sleepers. On the other hand, good sleepers showed a relatively lower percentage of energy from dietary carbohydrate and fat than poor sleepers. However, meta-regression analysis revealed no dose-dependent association between the macronutrient distributions and sleep duration. These results suggest that consuming a greater proportion of dietary protein may benefit on improving sleep quality in healthy adults. However, findings may be susceptible to reverse causality and additional RCTs are needed.

## 1. Introduction

The majority of our behavioural and physiological processes are mediated by the circadian system, which is a rhythm that repeats approximately every 24 h [1]. This biological clock is regulated by the suprachiasmatic nucleus (SCN) and is located in the anterior of the hypothalamus. Similar cells that made up these SCN can also be found in the peripheral cells, such as liver and intestine [2]. Together, these two “clocks” work in tandem to control numerous physiological processes such as food processing and energy homeostasis. This is achieved through the expression and activity regulation of enzymes involved in processes such as glucose, amino acid and cholesterol metabolism [3]. Disruption of the circadian coordination is often associated with hormone imbalance, increased risk of chronic diseases and reduced lifespan [4]. On the other hand, resetting this coordination was found to be beneficial to our overall well-being and increase in longevity [3].

One of the most commonly recognized circadian rhythm is the sleep-wake cycle [1]. The sleep-wake cycle allows an organism to synchronize with its environment. This maintains temporal organisation of its body’s endogenous processes which ultimately contribute to its health and survival potential [1]. An example is a mammal’s ability to anticipate periodic food availability. Restricted meal times were observed to induce behavioural and physiological anticipatory reactions, which were controlled by the circadian oscillator [5]. Conversely, food consumption, timed meals and a handful of nutrients can trigger a feedback loop to entrain this biological clock [3].

Today, insufficient sleep has become a growing global problem. In the last 40 years alone, the average sleep duration has decreased by two hours. This can be attributed to various factors, such as workload, lifestyle, social activities and technology [6]. Numerous studies have reported the negative impact of inadequate sleep on brain structure, activation and physiology [7]. A meta-analysis study found that short-term sleep deprivation has a varying impact on several cognitive domains: simple attention, complex attention, working memory, processing speed, short-term memory and reasoning [8]. The largest effect was observed for simple, sustained attention [8]. Apart from cognitive impact, growth hormones needed by the body for physical repair and renewal are also secreted during sleep. Sleep deprivation may lead to an impaired secretion of these hormones and compromise the body’s ability to repair itself and heal [9]. As a result, sleep disturbances, both quantitative and qualitative, may increase the risk of developing chronic conditions, such as obesity, metabolic disorders, diabetes and cardiovascular disease [6]. Chronic sleep deprivation was found to modify components of energy metabolism which may in turn alter the body’s metabolic function [1]. A crossover clinical study investigated the effects of 4-h restricted sleep over six consecutive days on healthy young men. During the period of sleep restriction, the subjects exhibited greater blood glucose levels and decreased insulin sensitivity upon ingestion of breakfast versus receiving 8 h of sleep [4]. The same study also observed a significant decrease in glucose disposal rate and decreased insulin secretion following an intravenous glucose tolerance test. This impaired glucose and insulin response was comparable to individuals who are aging and in early stages of type 2-diabetes [4]. Other experimental studies have revealed that sleep restriction may also lead to elevated hypercortisolaemia, increased C-reactive protein and other pro-inflammatory cytokine secretions, which may promote chronic inflammation [1]. 

Some methods that are usually recommended to improve sleep include eliminating external stimuli, such as loud noises and light, as well as implementing relaxation strategies that can be done through aromatherapy and massages [9]. Drug treatment is also often administered to aid sleep though its usage warrants caution due to potential side effects [10]. 

Nutritional intervention may be a healthier alternative over these extreme options. Studies in rodents have found that nutrients, such as glucose, amino acids and sodium can entrain the body’s circadian rhythm and hence influence sleep [11]. In different human populations, various traditional foods have been observed to have sleep promoting properties. Some examples are cow’s milk [12], chamomile tea [13], and tart cherries [14]. Unfortunately, human clinical evidence of the sleep-promoting effects of these foods are mostly conducted in small study populations. Hence, the findings remain unconfirmed and mechanism of action inconclusive [11]. 

Previous studies have reported that macronutrient intake is associated with sleep quality. It was found that large quantities of dietary carbohydrate (CHO) and dietary fat (FAT) may have the ability to modulate sleep quality by regulating the ratio of rapid eye movement (REM) and non-rapid eye movement (NREM) in sleep [15]. Dietary protein (PRO) may also influence sleep by supplying the body with amino acid tryptophan (TRP), which is a potential sleep-promoting nutrient [11]. TRP is first converted to 5-hydroxytryptophan (5-HTP), then to serotonin, which is then metabolized to melatonin to signal night time and the onset of sleep [16] A cross-sectional (CS) study has reported that the intake of PRO is positively correlated with sleep duration, quality and pattern [17]. However, studies on the effect of macronutrient distribution with sleep quality is limited and available results are inconsistent [18]. Therefore, the aim of this review is to systematically assess the association between sleep quality and macronutrient distribution in healthy adults by compiling and analyzing data from relevant published studies. Additionally, there is a significant association between sleep duration and obesity [19]. In this review, a sub-group analysis based on the subjects’ obesity status was also performed.

## 2. Materials and Methods 

### 2.1. Search Strategy

On 8 June 2018, a literature search was performed on PubMed, Medline (Ovid), CINAHL and COCHRANE database using a set of search terms (see Table A1). The search terms were structured to identify articles that examine the association of sleep quality and macronutrient distribution. An updated search was performed on 29 April 2019. 

### 2.2. Study Delection

Study selection of identified articles were decided by a set of inclusion criteria, adapted from a PICOS statement (see Table A2). The list of inclusion criteria included: (1) observational study and randomized controlled trial (RCT); (2) adults aged ≥ 19 years old (mean age); (3) collected daily macronutrient intake; and (4) conducted sleep quality assessment, such as sleep duration and global sleep score (GSS). Studies that reported evening meals and descriptive measures of sleep quality were excluded from this systematic review. Reviews and meta-analyses were also screened for any relevant articles not captured in the initial search. With the inclusion criteria, the identified articles were first screened by their titles and abstracts, followed by a full-text screening. The study selection flowchart is illustrated in Figure 1. The database search and study selection were performed by two reviewers (C.N.S. and W.M.X.) independently. Any discrepancies in the selection process were discussed until a consensus was reached.

### 2.3. Data Extraction

As part of the review process, the following data were also extracted by two reviewers (C.N.S. and W.M.X.) independently: year of publication, first author, title, study design, population characteristics, country of origin, gender, mean age, mean body mass index (BMI), subjects recruited, dietary data collection method, macronutrient energy distribution (E%), energy intake, and quantitative sleep quality measurements. As majority of our sleep occurs in the evening, both 24-h and nocturnal sleep duration were extracted from the studies. Additional information such as intervention duration, macronutrient distribution of the control and intervention diet were also obtained from the RCTs. Corresponding authors of the selected articles were contacted when additional data and clarification were needed. However, as no response was received, only published data were analysed in this review. 

### 2.4. Sleep Quality Analysis

Sleep quality was evaluated based on four main components: sleep duration, GSS, sleep latency (SL) and sleep efficiency (SE%). GSS is an arbitrary unit measurement obtained from the Pittsburgh Sleep Quality Index (PSQI). The PSQI is a self-rated questionnaire that assesses sleep quality for the past one month. It was designed to evaluate sleep quality in clinical populations and consists of 19 questions encompassing seven sleep components: subjective quality, SL, nocturnal sleep duration, SE%, sleep disturbances, the use of sleep medication and daytime dysfunction. The sum of the seven components would give a GSS score between 0 to 21, where a higher score indicates a poorer sleep quality [20]. 

In this review, a good sleeper was characterised as: (1) sleep duration ≥7 h; (2) GSS ≤ 5; (3) SL ≤ 30 min; (4) SE% ≥ 85%. On the other hand, sleep duration <7 h, GSS >5, SL >30 min and SE% <85%, are features of poor sleepers. These sleep quality parameters were based on recommended values by the National Sleep Foundation (NSF) and other previously published sleep studies [21]. The NSF has reported a recommended minimum of 7 h/day for young adults and older [21]. In addition, it has been reported that individuals with GSS >5 are clinically identified to have poor sleep quality [22]. A SE% <85% and SL >30 min on three or more nights in a week has also been defined to exceed normal clinical threshold and may suggest the presence of insomnia [23]. 

### 2.5. Dietary Analysis

The macronutrient distribution is reported in the form of percentage of energy intake (E%) of each macronutrient. These values were pooled together based on favourability of their designated sleep components: favourable (sleep duration ≥ 7 h; GSS ≤ 5; SL ≤ 30 min; SE% ≥ 85%) and non-favourable (sleep duration <7 h, GSS>5, SL>30 min and SE%<85%). The E% of each macronutrient of the different sleep components were then averaged to obtain the mean E%. These mean values were used for analysis to assess how macronutrients may influence sleep quality. Similarly, a sub-group analysis of the sleep duration’s macronutrient distribution data was also performed based on the obesity status of the recruited subjects. The obesity status was determined using the reported BMI and the World Health Organization (WHO) BMI guideline [24].

The Dietary Reference Intake (DRI) committee from the Institute of Medicine of the National Academies in the USA has established a macronutrient distribution guideline for a complete and healthy diet, which is the acceptable macronutrient distribution range (AMDR). The DRI committee has determined the AMDR to be 45–65%E CHO, 10–35%E PRO and 20–35%E FAT [25]. In addition to pooling the macronutrient distribution based on their sleep quality features, each of their means were compared with the AMDR. 

### 2.6. Quality Assessment

Quality Assessment Tool for Observational Cohort and Cross-Sectional Studies (QATOCC) and Quality Assessment of Controlled Intervention Studies (QACIS) were used to assess the quality of the corresponding selected studies. These assessment tools were developed by the National Heart, Lung and Blood Institute (NHLBI) to assess the risk of bias of research studies [26]. 

The quality of RCT studies in this review was assessed by the QACIS. RCT studies can attain a maximum score of 14 points, where its quality was classified by the following scores: 0–4 poor; 5–10 fair; and 11–14 good. 

On the other hand, quality assessment of observational studies was assessed using the QATOCC. However, due to the cross-sectional design of all selected studies, question 7 and 13 of the QATOCC were not applicable, as these questions are used to assess longitudinal studies. Therefore, unlike the RCTs, the CS studies could only attain a maximum score of 12 points. The distribution of the scores were: 0–4 poor; 5–8 fair; 9–12 good. 

Quality of assessment was done by two reviewers (C.N.S. and D.T.) to minimize bias. Any discrepancies in the quality assessment of the articles were discussed until a consensus was reached. 

### 2.7. Meta-Regression

Meta-regression analysis was conducted to assess the dose-dependent association between the individual macronutrient E% (CHO, PRO, FAT) and sleep quality. A random-effect model was applied using a univariable analysis. This meta-regression analysis was performed using Stata/IC (StataCorp LP, College Station, TX, USA) and statistical significance was accepted at *p* < 0.05.

## 3. Results

### 3.1. Study Characteristics

Nineteen articles, published between 2013 and 2018, met the selection criteria of this analysis. Fifteen of the studies were CS and four were RCTs (Figure 1). Table 1 and Table 2 summarized the characteristics of the selected CS and RCT studies, respectively. In the CS studies, most of the dietary information of their study participants were obtained through food frequency questionnaires (FFQ) and 24-h dietary recall. FFQ items applied in the selected studies ranges from 56 to 168 food items. From these nineteen selected articles, the sleep outcomes reported and extracted were sleep duration (17 articles), GSS from PSQI data (3 articles), SL (5 articles) and SE% (3 articles). As the majority of our sleep occurs in the evening, the sleep duration data extracted includes both 24-h and nocturnal sleep duration.

Overall, this systematic review included data collected from 86,961 individuals from eight different countries (seven studies from USA, three from Korea, two each from Australia, China, and Japan, and one each from Brazil, Iran, and Italy). Most of the participants recruited in these studies were also from generally healthy populations.

### 3.2. Macronutrient Distribution and Sleep Quality

The mean E% distribution of CHO, PRO and FAT based on their reported sleep quality characteristics were tabulated in Table 3. Data from the independent CS and RCT studies, as well as data from both study designs are also collectively provided in the same table.

In the sleep components sleep duration and GSS, the PRO E% of good sleepers from the combined CS and RCT data were 16.4% and 18.7%, respectively, while these were 15.9% and 15.9% in poor sleepers (Table 3). Although PRO E% from CS data were similar between good and poor sleepers in sleep duration and PSQI (15.7% vs. 15.3% and 13.6% vs. 13.4%), PRO E% of good sleepers in sleep duration, PSQI, SL and SE% were higher compared to poor sleepers (Good sleepers: 30.0%, 22.5%, 31.1% and 33.8% vs. poor sleepers: 18.3%, 16.7%, 20.0% and 20.0%) in RCT studies.

With the exception of sleep duration, good sleepers in GSS, SL and SE% had a lower FAT E% compared to the poor sleepers. The data from RCT studies of these three components, averaged 21.3%, 29.9% and 29.5% for good sleepers, while poor sleepers averaged on 38.3%, 31.3% and 31.3%, respectively. 

Similarly, good sleepers in sleep duration, SL and SE% had a lower CHO E% compared to poor sleepers. For these sleep components, RCT data were 45.0%, 39.4% and 37.5% for good sleepers and 52.5%, 48.8% and 48.8% for poor sleepers respectively. A similar trend was also observed in both the combined data and independent CS data for sleep duration. PSQI component from the RCT, on the other hand, seemed to favour a higher energy from CHO (56.3% vs. 45.0%). The combined mean data from PSQI also similarly gave a CHO E% average of 55.4% for good sleepers and 47.4% for poor sleepers. However, limited difference was observed in PSQI CS data (54.2% vs. 54.5%). 

A sub-group analysis based on the subject’s obesity status was performed and reported in Table 4. In the obese population, individuals with favourable sleep duration of ≥ 7 h displayed a higher mean E% from PRO and a lower E% from both CHO and FAT when compared to the poor sleepers of <7 h sleep. However, the macronutrient energy intake between the poor and good sleepers in non-obese population were similar.

### 3.3. Acceptable Macronutrient Distribution Range and Sleep Quality

Figure 2 compares the position of the RCT mean macronutrient distribution of poor and good sleeper within the AMDR. The same comparison from the combined mean (CS and RCT) and independent CS can be found in Figure A1 and Figure A2. 

From Figure 2, it can be observed that the mean of CHO, PRO and FAT E% from the collected studies were mostly within the AMDR. The CHO E% of good sleepers by sleep duration, GSS and SL was lower than poor sleepers. In addition to this, the mean of CHO E% for SL and SE% was below the AMDR range (Figure 2a). Although the mean of PRO E% for all sleep components was within the AMDR range, the PRO E% of good sleepers was higher than poor sleepers (Figure 2b). The mean FAT E% of all the good sleepers was within the AMDR while the mean FAT E% from GSS of poor sleepers was above the AMDR range (Figure 2c).

### 3.4. Quality Assessment 

The CS and RCT studies have an average quality assessment score of 7.2 (Table 1) and 10.3 (Table 2), respectively. All of the CS studies were of a fair quality (5–8). This is because their risk of detection bias was unclear, as the blinding of the assessors of these studies were not reported. In addition, data collection methods of most of the CS studies used were not as rigorous and this contributed to reporting bias. For the RCT studies, two of the studies were of a good quality (11–14) where they have been assessed to have a low risk of selection, detection, attrition and reporting biases while, the remaining two studies achieved a fair quality score (5–10). The lower score of these two studies were primarily contributed from the lack of blinding (subjects and/or assessor), which can give rise to some performance and detection biases in the study. However, all the RCT studies were adequately randomized, minimizing selection bias. A breakdown of the scoring allocation of the quality assessment of both the CS and RCT studies can be found in Table A3 and Table A4.

### 3.5. Meta-Regression

Meta-regression analysis was performed between the individual macronutrient distributions (CHO E%, PRO E% and FAT E%) and sleep duration data obtained from six of the selected articles of this systematic review [31,33,34,35,38,39]. A statistically significant dose-dependent association was not observed between this set of data (see Figure A3). 

A similar meta-regression analysis was not possible to be performed on the remaining set of sleep duration data and other sleep qualities: GSS, SL and SE%. This is due to the lack of available data required to run the meta-regression. 

## 4. Discussion

Sleep is mainly regulated by the light-dark cycle [1]. However, in addition to light, food can also be a powerful synchronizer (zeitgeber) of the sleep-wake cycle and this synchronization from external cues allows sleep to be carried out at appropriate times of the day [3]. Consumption of certain macro- and micronutrients and higher quality diet was reported to improve sleep quality and may have the ability to reset the body’s circadian rhythm [43]. Therefore, improving sleep through dietary changes seems to be a promising prospect. The association between macronutrients intake and sleep quality was observed in previous studies [11], and we also found that good sleepers (sleep duration ≥ 7 h; GSS ≤ 5; SL ≤ 30 min; SE% ≥ 85%), in general, followed dietary patterns with a higher energy distribution from PRO and relatively lower energy distribution from both CHO and FAT when compared to the poor sleepers (sleep duration < 7 h, GSS > 5, SL > 30 min and SE %< 85%).

A higher PRO diet has been studied to positively contribute to several health-related outcomes. Some of these outcomes include weight control and improving body composition [44]. Furthermore, this systematic review supports the evidence that a higher PRO diet improves sleep quality. A previous RCT study done by Zhou et al. [17] observed that a higher PRO intake improved the GSS score of research participants, which suggests better sleep quality [17]. In conjunction, an increased plasma TRP concentration was also observed in participants who consumed the higher PRO diet [17]. These results support that the potential mechanism for the impact of PRO on sleep may be related to the presence of tryptophan (TRP), which in turn impact the synthesis of serotonin and melatonin. These neurotransmitters are known to be involved in the sleep and wake cycle [17]. However, caution should be used, since obtaining TRP from a PRO-rich diet may not always be an effective way to induce sleep. Santana et al. [29] reported conflicting results, and showed a negative correlation between sleep and PRO intake [29]. A possible explanation could be that a high PRO diet may also contribute to the large neutral amino acids (LNAAs) concentration in the plasma. LNAAs include a group of non-polar amino acids which are valine, tyrosine, isoleucine, leucine and phenylalanine [45]. These LNAAs compete with TRP for the blood-brain barrier (BBB) transporter, hence potentially reducing the transport and availability of TRP into the brain, for its conversion to sleep-inducing neurotransmitters [17]. Another factor that may explain the inconsistency in this observation between different studies is the source of the protein. The availability of the TRP may vary depending on its source. One study by Zhou et al. [17] observed no difference in sleep quality between animal and plant-based protein. However, the main outcome of this RCT was not aimed to assess sleep quality [17]. Therefore, more studies are needed to assess different protein sources on its influence on sleep quality. 

Another possible way that PRO can induce sleep is through its effect on gut hormone secretion. As a zeitgeber, food availability can act as a signal to organize the body’s behavioural and physiological parameters. When food timing is restricted, mammals can exhibit anticipatory activity 2-3 h before meal time, and this is called food anticipatory activity (FAA). During FAA, variables such as body temperature, locomotion and gastrointestinal motility increases. In rats, FAA was found to alter sleep distribution. When FAA occurs, wakefulness increases, and the REM sleep pattern is altered [42]. Ghrelin is a hunger hormone which stimulate the increase in food intake. Fluctuations of this hormone was linked to food anticipation and may trigger an arousal. Intracerebroventricular injection of ghrelin in rats showed increase of wakefulness through NREM and REM suppression [46]. High PRO diet was reported to suppress ghrelin secretion [47], and this decreased ghrelin secretion may reduce FAA, and hence suppress its wake-promoting influence. Nevertheless, more research is required to confirm this speculation. Additionally, other than ghrelin, food ingestion is also regulated by other hormones, such as leptin, peptide-tyrosine-tyrosine (PYY) and glucagon-like-peptide 1 (GLP-1). Their role in food intake regulation may suggest their involvement in the food-entrainment mechanism, and possibly in the FAA [46]. Exploring their involvement in the FAA and sleep quality can be an invaluable information in the field of sleep nutrition and a prospective research direction. 

The findings in this review seemed to favour a lower energy distribution from FAT for better sleep quality; however, this finding is inconsistent with previous studies. A study by Rontoyanni et al. [48] on Greek adult female participants observed a positive association between sleep duration and saturated fat intake [48]. It was reported that a low-FAT diet was shown to reduce NREM sleep and increased REM sleep [15]. NREM is considered to be a deep sleep and a decrease in its proportion suggest a more unfavourable sleep architecture [15]. The type of FAT may play a role in its the sleep-inducing property. Santana et al. showed a negative correlation between sleep duration and intakes of monounsaturated fatty acids and cholesterol [29]. Long chain polyunsaturated fatty acids, such as arachidonic acid and oleic acid, were also reported to alter the rhythm of serotonin N-acetyltransferase (AANAT) activity and in turn may affect melatonin synthesis [49]. The sources of FAT used in the reviewed studies may not be rich in these unsaturated fatty acids, which are primarily in meat and dairy. Therefore, sources of FAT will need to be taken in consideration for future sleep studies. 

CHO was reported to increase postprandial insulin secretion, which mediates the uptake of LNAA, but not TRP, into the muscle. This insulin-mediated uptake into the muscle is not preferable to TRP because this amino acid is mainly bound to the plasma albumin, making it less bioavailable [11]. This in turn increases TRP concentration in the plasma and preferentially, assists its transport through the BBB which increases the availability of TRP to the brain for serotonin and melatonin conversion [17]. However, we observed conflicting results from this systematic review and this may be because of the wide variety of sugar chains that can be found in CHO. Its differences in structure may cause it to be metabolized differently, resulting in its inconsistent influence on sleep parameters [11]. For instance, Diethelm et al. [50] reported that high glycemic index CHO was accompanied by a longer sleep duration in toddlers [50]. In addition, Afaghi et al. [51] also reported that high glycemic index CHO diet was shown to significantly reduce SL [51]. In this systematic review we were unable to distinguish between high and low glycemic index of the CHO. This could be the source of inconsistencies in results observed in CHO impact on sleep. Observing the glycemic index degree of CHO and its extent to induce an insulin response could be a future direction on how CHO can influence sleep. In addition, PRO and CHO may work synergistically to improve brain TRP availability. Therefore, a comparison of PRO: CHO ratio in the future may be a beneficial analysis to elucidate further conclusions [41].

Since most of the data utilized in this study are collected from CS studies, caution should be exercised on the direction of the association and findings also can interpret as follow; people with better sleep quality may select a higher energy distribution from PRO and lower energy distribution from CHO or FAT in their diet. In childbearing aged women, it was observed that subjects who reported short sleep (~6 h) with sleeping difficulties and severe tiredness consumed higher percentage of energy as FAT [38]. Middle-aged Korean with short sleep durations (< 7 h) consumed more dietary CHO than those with long sleep durations (≥ 7 h) [52]. Previous studies have also reported that sleep restriction can reduce satiety hormones, leptin, while simultaneously increasing the level of ghrelin, which is an appetite inducing hormone [53]. Consequently, subjective hunger and appetite, especially for high CHO and high FAT foods, increased [54]. This may explain that those with poorer sleep quality may be more likely to consume higher amount of CHO and FAT and subsequently may decrease PRO E% distribution of their diet. In addition, very short sleepers (<5 h) [30] and insomniacs [43] reported lower PRO intake than normal sleepers. Though some RCT data has been reported in this study, a bidirectional association between macronutrients distribution and sleep quality may exist due to the nature of data collection therefore, more RCTs are required to further determine the causality and direction of the association.

In addition to the evaluation of the overall mean macronutrient E%, a sub-group analysis was performed on the sleep duration macronutrient data based on the subjects’ obesity status. Similarly, in obese subjects, those with favourable sleep duration also has a greater mean E% from PRO and lower mean E% from CHO and FAT. The macronutrient E% difference between the two sleep groups; however, was similar in non-obese subjects. Obesity has been considered as an important risk factor of sleep disorders, such as obstructive sleep apnea. The excessive body weight can act as a mechanical barrier that obstructs normal breathing which lead to frequent nocturnal awakening [55]. Previous studies has also reported that obese individuals were more likely to develop chronic insomnia [56]. Studies on the efficacy of L-TRP supplementation on sleep has reported that a higher dose of the supplement was required for individuals with severe chronic insomnia [57]. This may explain the higher E% PRO between good sleepers in the obese population as compared to the non-obese population. Obese subjects may benefit more from a diet with a higher energy E% from PRO, as a higher PRO consumption can provide more TRP, hence promoting longer sleep duration. 

Consumptions of macronutrients within AMDR has been associated with reduced risk of chronic diseases and most of the mean macronutrient distribution calculated for the components of sleep quality from this review also adhered to the AMDR [58]. In particular, mean PRO E% for sleep duration, GSS, SL and SE% of good sleepers was relatively higher than poor sleepers while the mean FAT E% from GSS of poor sleepers was above the AMDR range. Consistent with our findings, the study by Zhou et al. [17] reported that middle-aged adults who consumed a diet with 20 or 30 E% PRO improved a sleep quality compared to a diet with 10 E% PRO [17]. Thus, this study proposes a PRO E% range within the AMDR for good sleep quality. Increasing the PRO E% to the higher end of the AMDR may have the added benefit of improving sleep quality, while still maintaining the recommended macronutrient range that has been suggested to reduce risk of chronic diseases. 

Although sleep duration is one of the important components of determining sleep quality [59] the restorative benefit of sleep is also determined by its quality, architecture and timing [21]. The main strength of this review is the inclusion of a variety of sleep components: sleep duration, GSS, SL and SE%. This not only allows a more streamlined observation, but also a more comprehensive assessment of how macronutrient distribution can be associated with the different components of sleep quality. In addition, comparing the macronutrients distribution in good and poor sleepers with the currently recommended macronutrient distribution allows us to incorporate the use of AMDR as a dietary recommendation to also encourage healthy sleep patterns. However, this review used the US AMDR as the reference for macronutrient range comparison. Although the AMDR used by other countries, such as Australia [60], Japan [61] China [62] and Korea [63], have similar macronutrient distribution, cultural differences still need to be considered when future dietary suggestions between cultures are made. 

We restricted this study to articles published in English only, and this may limit our study selection. Due to the paucity of the data, statistical analysis was restricted. The meta-regression analysis performed to assess the dose-dependent relationship was only possible on a limited number of data sets analysed in this review. The analysis, in turn, did not observe any statistically significant association between the macronutrient E% and sleep duration. Additional studies are required to acquire a greater data set for a more extensive statistical analysis on sleep and nutrition. In addition, data used in this systematic review were mostly collected from CS studies, and most of the observations made in this review are associative in nature. Moreover, caution should be used in the interpretation of this association, since the direction of the association cannot be assessed in this review and there may be a bidirectional association between macronutrients and sleep. Another weakness of this study is the variety of methods that was used across the CS studies to collect dietary data. The use of different dietary collection tool (food record vs. frequency questionnaire), can lead to inconsistency in observed results. No single dietary collection method is perfect, and each has its own limitations [64]. 

Sleep and nutrition research is still in its infancy, and it offers numerous unexplored research opportunities. Some studies have suggested that the timing and content of meals can also influence sleep [65]. Studies observing the impact of the macronutrient composition of a single meal, such as evening meals, can provide invaluable information in the area of sleep nutrition. For example, a study observed that a CHO-rich evening meal seemed to reduce NREM sleep when compared to its FAT-rich meal comparison [66]. Additionally, although lack of sleep is detrimental to health, excessive sleep is also not ideal. It has been reported that long sleep duration (≥ 9 h) may also be associated with poor physical and mental health [30]. However, this was not assessed in this review. Thus, to better understand and develop strategies in improving sleep quality through diet, excessive sleep duration will also need to be considered as poor sleep in subsequent analysis. In the future, this research field can benefit from more rigorous studies to provide additional insight on the utilisation of nutrition to improve sleep quality.

## 5. Conclusions

This systematic review suggests that consuming a greater proportion of energy from dietary protein may benefit on improving sleep quality in healthy adults. However, a reverse causality may be susceptible and more RCT data are required to confirm this causality. When confirmed, these findings may provide a scientific background and research direction for the use of dietary recommendations for the management of macronutrient distribution to improve sleep quality in healthy adults by healthcare professionals. 

## Figures and Tables

**Figure 1 nutrients-12-00126-f001:**
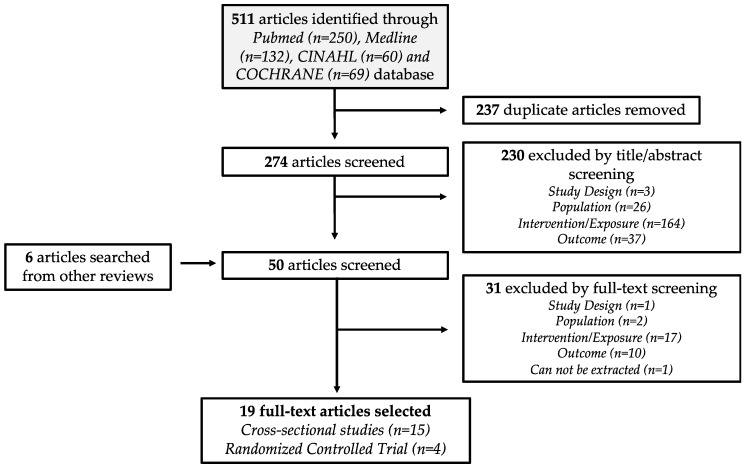
Flow diagram of the identification and selection of relevant studies.

**Figure 2 nutrients-12-00126-f002:**
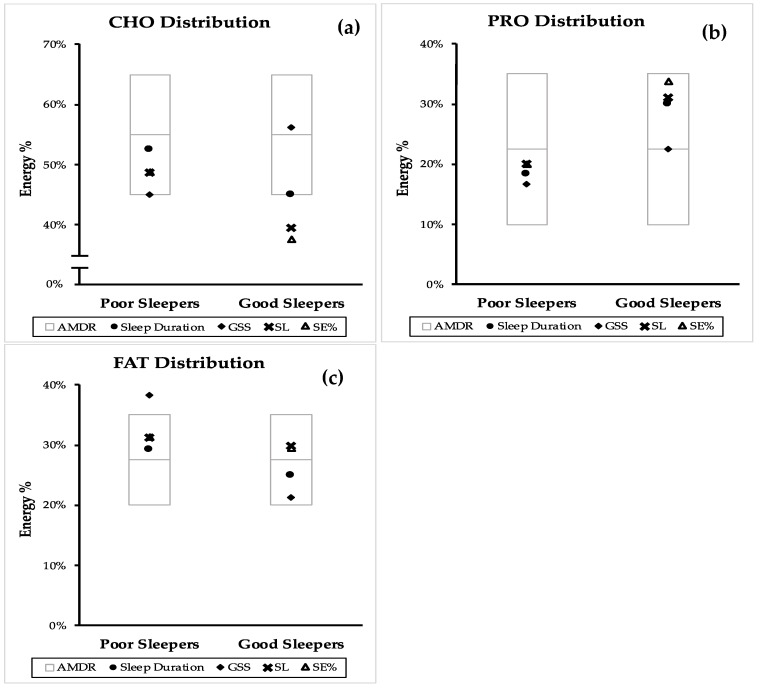
Comparison of good and poor sleep macronutrient distribution with the accepted macronutrient distribution (AMDR)* from randomized controlled trials (RCT)**.** Macronutrient distribution are presented as mean energy percentage values generated from sleep duration, PSQI, sleep latency (SL) and sleep efficiency (SE%). (**a**) dietary carbohydrate (CHO) distribution, (**b**) dietary protein (PRO) distribution and (**c**) dietary fat (FAT) distribution. AMDR is presented as area within the box plot.

**Table 1 nutrients-12-00126-t001:** Study design and subject characteristics of selected cross-sectional (CS) studies on the association between macronutrient distribution and sleep quality.

Citation	Country	Population	*n*	Age [mean, SD](y)	BMI [mean, SD]	Dietary Collection Method	Measurement of Sleep Quality	Quality Assessment Score
Shi et al., 2008 ^1^ [27]	China	Healthy	2828	47.1 (14.2)	23.5 (3.5)	3-Day food record	Sleep duration (24-h)	8
Haghighatdoost et al., 2012 [28]	Iran	Healthy	223	20.7 (1.8)	22.5 (2.8)	FFQ	Sleep duration (Nocturnal)	6
Santana et al., 2012 [29]	Brazil	Elderly and obese	58	66.2 (4.0)	33.8 (2.7)	24-h Recall	Sleep duration (Nocturnal)	5
Grandner et al., 2013 ^2^ [30]	USA	Healthy	4548	46.3 (16.5)	28.7 (6.8)	24-h Recall	Sleep duration (Nocturnal)	8
Kant & Graubard, 2014 ^2^ [31]	USA	Healthy	15,199	-	-	24-h Recall	Sleep duration (Nocturnal)	6
Katagiri et al., 2014 [32]	Japan	Healthy	3129	47.6 (4.0)	21.9 (3.0)	DHQ	Sleep duration (Nocturnal), PSQI	8
Poggiogalle et al., 2016 [33]	Italy	Healthy (Caucasian)	42	52.5 (11.2)	40.1 (6.7)	3-Day food record	Sleep duration (24-h)	8
Doo & Kim, 2016 ^3^ [34]	Korea	Healthy	14,111	44.8 (0.3)	23.7 (0.1)	FFQ	Sleep duration (24-h)	8
Doo et al., 2016 ^3^ [35]	Korea	Healthy	14,680	45.2 (0.3)	23.6 (0.1)	FFQ	Sleep duration (24-h)	8
Heath et al., 2016 [36]	Australia	Healthy (Shift-workers)	118	43.4 (9.9)	27.1 (4.3)	FFQ	Sleep duration (24-h)	8
Doo & Kim, 2016 ^3^ [37]	Korea	Healthy	14,111	44.9 (0.2)	23.7 (0.1)	FFQ	Sleep duration (24-h)	8
Bennett et al., 2017 [38]	Australia	Healthy	6594	33.7	26.0	FFQ	Sleep duration (24-h)	8
Spaeth et al., 2017 [39]	USA	Healthy	46	33.9 (9.1)	24.5 (3.6)	Weight	Sleep duration (Nocturnal), SL, SE%	6
Komada et al., 2017 [40]	Japan	Healthy	1902	48.0 (10.3)	22.4 (3.3)	BDHQ	Sleep duration (Nocturnal), SL	7
Liu et al., 2018 ^1^ [41]	China	Healthy	9239	50.5 (15.0)	23.35 (3.5)	24-h Recall	Sleep duration (24-h)	6
							**Average Score**	**7.2**

^1^ Data obtained from China Health and Nutrition Survey (CHNS), ^2^ Data obtained from National Health and Nutrition Examination Survey (NHANES), ^3^ Data obtained from the Korean National Health and Nutrition Examination Survey (KNHANES), BDHQ (brief diet history questionnaire); BMI (body mass index); DHQ (dietary history questionnaires); FFQ (food frequency questionnaire); PSQI (Pittsburgh sleep quality index); SL (sleep latency); SE% (sleep efficiency).

**Table 2 nutrients-12-00126-t002:** Study design and characteristic of selected randomized controlled trial (RCT) on the impact of macronutrient distribution on sleep quality.

Citation	Study Design	Country	Population	*n*	Age [mean, SD] (y)	BMI [mean, SD]	Control	Intervention	Intervention Duration	Sleep Evaluation	Quality Assessment Score
Lindseth et al., 2013 [41]	Double-blind Crossover	USA	Healthy	44	20.6 (2.0)	24.8 (3.5)	35% FAT 50% CHO 15% PRO	**High-FAT** (**56% FAT**, 22% CHO, 22% PRO) **High-CHO** (22% FAT, **56% CHO**, 22% PRO) **High-PRO** (22% FAT, 22% CHO, **56% PRO**)	4 days (2-wk washout)	SL, SE%	12
Karl et al., 2015 [42]	Block	USA	Healthy	39	21.0 (3.7)	25.0 (3.7)	31% FAT 55% CHO 14% PRO	**Moderate-PRO** [30% FAT, 43%CHO, 27%PRO] **High-PRO** [30%FAT, 28%CHO, 42%PRO] ***40% energy deficit**	21 days	SL	7
Lindseth & Murray, 2016 [18]	Crossover	USA	Healthy	36	20.9 (1.9)	24.6 (4.1)	35% FAT 50% CHO 15% PRO)	**High-FAT** (65% **FAT**, 25% CHO, 10% PRO) **High-CHO** (10% FAT, 80% **CHO**, 10% PRO) **High-PRO** (15% FAT, 40% CHO, 45% **PRO**)	4 days (2-wk washout)	Sleep duration (*Nocturnal*), PSQI, SL, SE%	12
Zhou et al., 2016 [17]	Crossover	USA	Overweight/Obese	14	56.0 (3.0)	30.9 (0.6)	25% FAT 65% CHO 10% PRO	**Moderate-PRO** (25% FAT, 55% CHO, **20% PRO**) **High-PRO** (25% FAT, 45% CHO, **30% PRO**) ***750kcal energy deficit**	4 weeks	Sleep duration (*Nocturnal*), PSQI	10
										**Average Score**	**10.3**

BMI (body mass index); CHO (dietary carbohydrate), FAT (dietary fat); PRO (dietary protein); PSQI (Pittsburgh sleep quality index); SL (sleep latency); SE% (sleep efficiency). *Intervention diet included an energy deficit from the subjects’ normal diet.

**Table 3 nutrients-12-00126-t003:** Macronutrient distribution (E%) comparison between poor and good sleepers based on sleep duration, GSS, SL and SE% value.

Data Groups		POOR SLEEPERS			GOOD SLEEPERS		
		CHO	PRO	FAT	CHO	PRO	FAT
**SLEEP DURATION**		**< 7 h duration**			**≥ 7 h duration**		
**Combined**	**Mean (E%)**	55.0	15.9	27.6	52.1	16.4	30.0
	Range (E%)	25.0–80.0	10.0–45.0	10.0–65.0	39.1–67.2	12.2–30.0	18.4–36.1
**CS**	**Mean (E%)**	55.6	15.3	27.3	52.5	15.7	30.2
	Range (E%)	38.7–69.9	12.8–21.5	16.3–36.8	39.1–67.2	12.2–20.7	18.4–36.1
**RCT**	**Mean (E%)**	52.5	18.3	29.2	45.0	30.0	25.0
	Range (E%)	25.0–80.0	10.0–45.0	10.0–65.0	45.0–45.0	30.0–30.0	25.0v25.0
**PSQI**		**GSS > 5**			**GSS ≤ 5**		
**Combined**	**Mean (E%)**	47.4	15.9	36.0	55.4	18.7	24.6
	Range (E%)	25.0–65.0	10.0–30.0	25.0–65.0	40.0–80.0	10.0–45.0	10.0–35.0
**CS**	**Mean (E%)**	54.5	13.4	29.0	54.2	13.6	29.1
	Range (E%)	54.4–54.5	13.4–13.4	29.0–29.0	53.8–54.5	13.6–13.7	28.8–29.5
**RCT**	**Mean (E%)**	45.0	16.7	38.3	56.3	22.5	21.3
	Range (E%)	25.0–65.0	10.0–30.0	25.0–65.0	40.0–80.0	10.0–45.0	10.0–35.0
**SLEEP LATENCY (SL) ***		**SL > 30 min**			**SL ≤ 30 min**		
**RCT**	**Mean (E%)**	48.8	20.0	31.3	39.4	31.1	29.9
	Range (E%)	25.0–80.0	10.0–45.0	10.0–65.0	22.0–56.0	14.0–56.0	18.0–56.0
**SLEEP EFFICIENCY (SE%) ***		**SE% ≤ 85%**			**SE% > 85%**		
**RCT**	**Mean (E%)**	48.8	20.0	31.3	37.5	33.8	29.5
	Range (E%)	25.0–80.0	10.0–45.0	10.0–65.0	22.0–56.0	22.0–56	18.0–56.0

* Combined and CS data not available for SL and SE%.

**Table 4 nutrients-12-00126-t004:** Macronutrient distribution (E%) comparison between poor and good sleepers based on sleep duration and obesity status.

Data Groups		POOR SLEEPERS			GOOD SLEEPERS		
		CHO	PRO	FAT	CHO	PRO	FAT
**SLEEP DURATION**		**<7 h duration**			**≥ 7 h duration**		
**Non-obese ***	**Mean (E%)**	55.2	15.6	27.5	52.9	15.7	29.9
	Range (E%)	25.0–80.0	10.0–45.0	10.0–65.0	39.1–67.2	12.2–20.7	18.4–36.1
**Obese* ***	**Mean (E%)**	52.7	17.9	28.3	42.2	22.2	32.1
	Range (E%)	38.7–65.0	10.0–21.5	24.9–34.5	40.7–45.0	19.0–30.0	25.0–35.9

*** Non-obese: BMI<30 (General population); BMI<27.5 (Asian), ** Obese: BMI ≥30(General population); BMI ≥ 27.5 (Asian).

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
