# Peer review of "Association of Sleep Quality and Macronutrient Distribution: A Systematic Review and Meta-Regression"

_nutrients, 2020, doi:10.3390/nu12010126_

Round 1

Reviewer 1 Report

3 December 2019

To the authors:

This systematic review is of potential importance because it investigates a knowledge gap about sleep and macronutrients. It is very well written. However, one key issue is the lack of discussion of a potential bidirectional relationship between diet and sleep. There are a few other issues that need attention, and it’s possible that they were addressed and just not stated in the text. If that is the case, more transparency is necessary.

Main points:

Title:

Why does the title not include that it is a meta-regression analysis?

Abstract:

It should be noted in the abstract that the focus was healthy adults. Also, the meta-regression analysis should be noted there.

I’m not sure you can go so far as to state, in line 21, that dietary protein affects sleep as most included studies were cross sectional, and therefore directionality cannot be commented on.

Introduction:

Again, please note your age group. For example, on line 103, at the end of the sentence ending in “…published studies”, you could add “in heathy adults”.

Line 104 – As it introduces bias in a systematic review to state a hypothesis, please remove this statement.

Materials and methods:

Please include the day, month and year of each search. Currently only month and year are shown.

Study selection – Why were cohort studies not included? This seems odd and if they were excluded from the search criteria, what is the reason? It should be spelled out explicitly if in fact they were purposely excluded, and it should be a stated limitation of the study.

Data extraction – it is not clear that your data extraction was done independently by two reviewers. If it was, please state explicitly. If it was not, this should be listed as a limitation.

Quality assessment – this section is too large and I would cut most of the detail. If space is an issue for including other things I’ve mentioned, this would be the place to save space.

Results:

In the Results section line 218 – it would be helpful to know the range of item numbers in the included FFQs.

In section 3.5, re: meta-regression, line 330 relates that some analysis was not possible. I assume this is due to heterogeneity of outcomes or measurement tools, but please state this.

Discussion - Limitations:

It is not enough to say that the inclusion of cross sectional studies made your observations associative. The fact that the meta-analysis included cross sectional studies prevents any understanding of the direction of the relationship discussed, and this should be noted as a limitation. Thus, it is important to state that there may be a bidirectional relationship between macronutrients and sleep that cannot be assessed here, and point to what should be done in the future.

According to Table A1, English was the only included language. If this is the case it must be listed as a limitation.

Excluding cohort studies should be listed as a limitation.

If data extraction was not completed by two reviewers, this should be listed as a limitation.

Conclusions:

Again, please state these findings are in healthy adults.

As stated earlier, you cannot state anything about directionality. There may be a bidirectional relationship between dietary protein and sleep.

Other points:

Referencing issues: Intro, line 65: The sentence ending in “reasoning” requires a reference.

Methods, line 153: The sentence discussing recommendations by the NSF requires a reference.

Methods, line 179: the first sentence of section 2.6 introducing the tool used requires a reference.

Discussion, lines 340-344: this sentence requires referencing.

Minor things:

Results, quality assessment, line 268: should be a paragraph indent.

Figure 1 – not crazy about the data visualization as it’s hard to see the symbols that are superimposed but at this late date it’s probably hard to change.

Discussion, line 355: the full stop should go after “et al” not the ref number.

Discussion, line 384-385: word choice – “A study by Rontoyannie, et al. (48) on Greek women observed …” would be better as “A study by Rontoyannie, et al. (48) in Greek adult female participants…”

Discussion, line 439: “streamline” should be “streamlined”.

Author Response

Dear Editors and Reviewers,

Thank you for taking the time to provide us with further suggestions to help us improve our manuscript. Our responses are as follows.

Reviewer 1:

Title:

Why does the title not include that it is a meta-regression analysis?

As suggested, we included “meta-regression analysis” in the title.

Abstract:

It should be noted in the abstract that the focus was healthy adults. Also, the meta-regression analysis should be noted there.

We appreciate the reviewer’s significant comment and we made a revision of the conclusion statement in the abstract. Please see line 22-25. We also agree with reviewer’s suggestion regarding “healthy adults” and now they are added in lines 16 and 21-22.

“In this study we aim to assess the association between sleep quality and macronutrient distribution in healthy adults from systematically reviewed cross-sectional studies and randomized controlled trials.”

“However, meta-regression analysis revealed no dose-dependent association between the macronutrient distributions and sleep duration.”

I’m not sure you can go so far as to state, in line 21, that dietary protein affects sleep as most included studies were cross sectional, and therefore directionality cannot be commented on.

As we observed a higher energy distribution from dietary protein in good sleepers than poor sleepers in both randomized controlled trials and cross-sectional studies, we will maintain the current conclusion statement. However, we have elaborated this point in the conclusion section of the manuscript and our further response to your later comment in conclusion is made below. We appreciate the reviewer’s significant comment.

Introduction:

Again, please note your age group. For example, on line 103, at the end of the sentence ending in “…published studies”, you could add “in heathy adults”.

Now “in healthy adults” is noted in line 104.

“Therefore, the aim of this review is to systematically assess the association between sleep quality and macronutrient distribution in healthy adults by compiling and analysing data from relevant published studies.”

Line 104 – As it introduces bias in a systematic review to state a hypothesis, please remove this statement.

Thanks for your suggestion and now we removed the hypothesis statement.

Materials and methods:

Please include the day, month and year of each search. Currently only month and year are shown.

An initial search and an updated search were performed on 8th June, 2018 and 29th April, 2019, respectively and we have included this information in lines 110 and 113. Thanks for your kind suggestion.

Study selection – Why were cohort studies not included? This seems odd and if they were excluded from the search criteria, what is the reason? It should be spelled out explicitly if in fact they were purposely excluded, and it should be a stated limitation of the study.

In our PICOS statement (Table A2), we originally indicated observational study and randomized controlled trial as our study setting and we did not purposefully exclude cohort studies for the study selection. Cohort studies was not captured in the initial search of this study. Studies excluded based on study design are reviews or study protocols. However, we found that we inadvertently specified cross-sectional (CS) study as one of inclusion criteria in line 117, therefore revision has been made accordingly. We appreciate you catching this inadvertent mistake.

Data extraction – it is not clear that your data extraction was done independently by two reviewers. If it was, please state explicitly. If it was not, this should be listed as a limitation.

Data extraction was also done by two reviewers independently and as you suggested, we included this information in lines 132-133.

“As part of the review process, the following data were also extracted by two reviewers (C.N.S. and W.M.X.) independently:”

Quality assessment – this section is too large and I would cut most of the detail. If space is an issue for including other things I’ve mentioned, this would be the place to save space.

Thank you for your suggestion and quality assessment section has been edited as advised.

Results:

In the Results section line 218 – it would be helpful to know the range of item numbers in the included FFQs.

Thank you for your suggestion. The range of item numbers of the FFQ has been added in line 206.  

“FFQ items applied in the selected studies ranges from 56 to 168 food items”

In section 3.5, re: meta-regression, line 330 relates that some analysis was not possible. I assume this is due to heterogeneity of outcomes or measurement tools, but please state this.

To run meta-regression analysis, outcomes should include mean values of the outcome and their standard deviation. However, these information was not made available by the authors hence we were not able to run the analysis. To clarify this, a statement on why some of the meta-regression was not ran has been added in the manuscript (line 318-319). Thank you for your suggestion.

“This is due to the lack of available data required to run the meta-regression.”

Discussion - Limitations:

It is not enough to say that the inclusion of cross sectional studies made your observations associative. The fact that the meta-analysis included cross sectional studies prevents any understanding of the direction of the relationship discussed, and this should be noted as a limitation. Thus, it is important to state that there may be a bidirectional relationship between macronutrients and sleep that cannot be assessed here, and point to what should be done in the future.

We appreciate the feedback and concur with the reviewer’s comment. We highlighted the caution of the interpretation with a potential bidirectional association and suggestion has been made in lines 443-446.

“Moreover, caution should be noted in the interpretation of this association since the direction of the association cannot be assessed in this review and there may be a bidirectional association between macronutrients and sleep. Therefore, more available data from RCT are required which provide the causality and direction of the association.”

According to Table A1, English was the only included language. If this is the case it must be listed as a limitation.

Thank you for your comment and this is considered as limitation in line 436.

“We restricted to articles published with English only and this may limit our study selection.”

Excluding cohort studies should be listed as a limitation.

As we described above, both of observational study and randomized controlled trial were our study setting and we did not intentionally exclude cohort studies for the study selection. During the study selection process, we were unable to capture cohort studies.

If data extraction was not completed by two reviewers, this should be listed as a limitation.

As we indicated above, data extraction was done by two reviewers independently.

Conclusions:

Again, please state these findings are in healthy adults.

Now “in healthy adults” is noted in line 467.

“The findings from this systematic review suggest that consuming a greater proportion of energy from dietary protein may have a beneficial influence on sleep quality in healthy adults.”

As stated earlier, you cannot state anything about directionality. There may be a bidirectional relationship between dietary protein and sleep.

Although we observed a higher energy distribution from dietary protein in good sleepers than poor sleepers in both randomized controlled trials and cross-sectional studies, we also agree with the reviewer’s valuable comment and we have elaborated the possible a bidirectional association between macronutrients and sleep in line 465-469.

“While a reverse causality may be susceptible, the findings from this systematic review suggest that consuming a greater proportion of energy from dietary protein is associated with better sleep quality in healthy adults. These findings also provide a strong background for healthcare professionals when dietary recommendations for the management of macronutrient distribution to improve sleep quality in healthy adults. Nevertheless, more RCT data are required to confirm them.”

Other points:

Referencing issues: Intro, line 65: The sentence ending in “reasoning” requires a reference.

Reference has been added to the sentence as suggested.

Methods, line 153: The sentence discussing recommendations by the NSF requires a reference.

Reference has been added to the sentence.

Methods, line 179: the first sentence of section 2.6 introducing the tool used requires a reference.

Reference of the assessment tool has been added.

Discussion, lines 340-344: this sentence requires referencing.

Reference regarding previous studies on macronutrient and sleep quality has been added as suggested. However, the second part of the sentence is a result summary based on the finding of this study. As it is used as an opening paragraph for the discussion section, reference was not added.

Minor things:

Results, quality assessment, line 268: should be a paragraph indent.

Thank you for the advice. As requested, the changes have been made accordingly.

Figure 1 – not crazy about the data visualization as it’s hard to see the symbols that are superimposed but at this late date it’s probably hard to change.

May we confirm that the figure reviewer is referring to is “Figure 2” and not “Figure1”. We appreciate our reviewer’s feedback on the Figures used in this manuscript. However, as the purpose of Figure 2 is to visualise the distribution of each of mean macronutrient E% with reference to the AMDR, we have decided to keep it as it is. The actual mean values macronutrient E% has already been reported in Table 3 (Line 266)

Discussion, line 355: the full stop should go after “et al” not the ref number.

Thank you for your suggestion, referencing has been edited.

Discussion, line 384-385: word choice – “A study by Rontoyannie, et al. (48) on Greek women observed …” would be better as “A study by Rontoyannie, et al. (48) in Greek adult female participants…”

Thank you for the advice and the change has been made accordingly.

Discussion, line 439: “streamline” should be “streamlined”.

We appreciate you catching this inadvertent mistake and conducted English editing.

Reviewer 2 Report

The systematic review by Sutanto et al. titled Association of sleep quality and macronutrient distribution: a systematic review aims to systematically assess the association between sleep quality and macronutrient distribution by compiling and analysing data from relevant published studies.

The study covers this interesting research area and has been through the review process before. It is well-written and clear and it seems that issues raised by previous reviewers have been addressed. I have some minor comments:

Results:

The text referring to Figure 1 is somewhat lacking in information. Please provide. The text refers to a Figure 2 but this is not included in the manuscript as far as I can see. Please provide.

Author Response

Dear Editors and Reviewers,

Thank you for taking the time to provide us with further suggestions to help us improve our manuscript. Our responses are as follows.

Reviewer 2:

The text referring to Figure 1 is somewhat lacking in information. Please provide.

Thank you for your suggestion. “Study selection flow diagram” description of Figure 1 has been paraphrase to “Flow diagram of the identification and selection of relevant studies”

The text refers to a Figure 2 but this is not included in the manuscript as far as I can see. Please provide.

The text referring to Figure 2 (line 295) can be found at line 243-252.

This manuscript is a resubmission of an earlier submission. The following is a list of the peer review reports and author responses from that submission.

Round 1

Reviewer 1 Report

This is an interesting systematic review that contributes to build evidence in this field. The objectives are clear, the methods adequate and appropriately described. The tables are clear.

Why did the authors excluded from analysis studies reporting evening meals? Do you mean ONLY evening meals?

Were the results different depending on the dietary assessment tool used in the studies?

Why did the authors did not run meta-analysis? This will benefit the paper.

English should be checked for a few spelling / grammar errors.

Reviewer 2 Report

This is a very interesting review  on the impact of macronutrient distribution on sleep health. This has the potential to inform recommendations for macronutrient intake and improve the crisis of inadequate sleep. The authors have done a good job of searching and analysing the literature. There are some errors with spelling and grammar that need to be addressed. Additionally, I have made some minor suggestions below to help the argument of the manuscript. 

Introduction

Line 52 - include more description of the sleep/wake cycle and what it promotes (awake during the day and asleep at night) since disrupted sleep is a major part of this paper. It would also be good to include more description of the 24h cycle in food, that is, we are primed to digest food during the day and not at night Line 56 - change "sleep deprivation" to "inadequate sleep" Line 59 - include some brief discussion on how cognitive performance is impacted by poor sleep, as this is a major reason why it is good to get adequate sleep and why people may benefit from a macronutrient intervention to improve sleep.  Line 67 - include "be" between "can" and "characterized" Line 73 - change "reset" to "entrain" 

Methods

Line 104-105 - why were subjective sleep measures excluded but subjective measures of macronutrients (eg 24h recall) were included? Some explanation of this would be useful.  Line 147-152 - does the AMDR take into account cultural differences in macronutrient intake? If yes, then this should be discussed, if no then this should be considered as the dietary recommendations and the suggestions for protein/fat/carbohydrate intake may differ between cultures. 

Results

Table 1 - What is the justification for including the Heath et al study with a shiftwork sample, given that sleep and macronutrient intake in this sample is likely to be effected by shiftwork and different interventions may be required compared to traditional workers with diurnal schedules? Table 3 - could you label the Mean (%) and Range (%) as Mean (E%) and Range (E%) so it is clear what the values are referring to

Discussion

Lines 280-292 - this is really good information and provides a good context for why this literature review is important. Consider moving this to the introduction to provide greater background. 

Reviewer 3 Report

The present study carried out by Sutanto et al. investigated associations between sleep quality and macronutrient distribution. In this systematic review of cross-sectional studies and randomized controlled trials, they selected 19 relevant articles. Based on the PSQI questionnaire, authors assessed four subjective criteria of sleep quality - sleep duration hours, global sleep score, sleep latency and sleep efficiency. Authors found that protein intake was higher and fat and carbohydrate intake lower in good sleepers compared with poor sleepers. They concluded that consuming a greater proportion of energy from dietary protein may have a beneficial influence on sleep quality.

All in all, the manuscript raises an interesting question whether sleep quality is associated with macronutrient intake. However, the manuscript has a range of gaps and limitations.

The PSQI questionnaire allows a subjective assessment of a sleep quality. Why is it mentioned that “studies… only subjective sleep quality were excluded from review” ? (page 3, line 104) Did authors analysed studies use objective criteria of the sleep quality like EEG? There is only one old work from the year 1975 cited (line 82). Please include the comparison of studies with objective and subjective sleep criteria concerning macronutrient distribution at least in the discussion. There is no statistical analysis for any difference between groups of good and poor sleepers. Please perform the meta-analysis of your data and include them in the manuscript. There are many factors which affect sleep quality such as caffeine, nicotine, and alcohol consumption, stress and anxiety, metabolic diseases etc. Did you consider these factors to avoid misinterpretation? In particular, an interesting question is whether obese and non-obese subjects show the same results concerning sleep quality and macronutrient distribution. Please add these data to the manuscript. Next, dietary protein has differed amino acid composition including tryptophan amounts depending on its origin (e.g. plant and animal protein). These have to be considered in the analysis. AMDR for protein have a great range between 10% and 35% and therefore cannot be used as a reference point. Line 74: Please exchange the phrase “specific food” with “traditional food” and cite original papers to this issue except of a review. Line 325: High protein intake affects not only ghrelin levels but also secretion of insulin, glucagon and other postprandial hormones. How they affect arousal, satiety and sleep quality? Again – are there differences in protein-induced hormone secretion in metabolically healthy and ill humans? Figures are often shifted and the title of the figure A1 is not readable. Please explain which quality criteria are designated as Q1-14.